# A Surface-Enhanced Raman Spectroscopy-Based Aptasensor for the Detection of Deoxynivalenol and T-2 Mycotoxins

**DOI:** 10.3390/ijms25179534

**Published:** 2024-09-02

**Authors:** Rugiya Alieva, Svetlana Sokolova, Natalia Zhemchuzhina, Dmitrii Pankin, Anastasia Povolotckaia, Vasiliy Novikov, Sergey Kuznetsov, Anatoly Gulyaev, Maksim Moskovskiy, Elena Zavyalova

**Affiliations:** 1Chemistry Department of Lomonosov Moscow State University, Moscow 119991, Russia; ruqiwa_eva@mail.ru (R.A.); svetlanasokolova02@mail.ru (S.S.); 2All-Russian Research Institute of Phytopathology, Bolshiye Vyazemy 143050, Russia; zhemch@mail.ru; 3Center for Optical and Laser Materials Research, St. Petersburg State University, St. Petersburg 199034, Russiaanastasia.povolotckaia@spbu.ru (A.P.); 4Federal Scientific Agroengineering Center VIM, Moscow 109428, Russia; vs.novikov@kapella.gpi.ru (V.N.); kuznetsovsm@kapella.gpi.ru (S.K.); tomasss1086@mail.ru (A.G.); maxmoskovsky74@yandex.ru (M.M.); 5Prokhorov General Physics Institute of the Russian Academy of Sciences, Moscow 119991, Russia

**Keywords:** aptamer, aptasensor, SERS, mycotoxin, DON

## Abstract

The quality of food is one of the emergent points worldwide. Many microorganisms produce toxins that are harmful for human and animal health. In particular, mycotoxins from Fusarium fungi are strictly controlled in cereals. Simple and robust biosensors are necessary for ‘in field’ control of the crops and processed products. Nucleic acid-based sensors (aptasensors) offer a new era of point-of-care devices with excellent stability and limits of detection for a variety of analytes. Here we report the development of a surface-enhanced Raman spectroscopy (SERS)-based aptasensor for the detection of T-2 and deoxynivalenol in wheat grains. The aptasensor was able to detect as low as 0.17% of pathogen fungi in the wheat grains. The portable devices, inexpensive SERS substrate, and short analysis time encourage further implementation of the aptasensors outside of highly equipped laboratories.

## 1. Introduction

The world population is expected to grow by over a third, from 6.8 billion people in 2009 to 9.1 billion people in 2050, an increase of 2.3 billion people. Nearly all of this population growth is forecast to take place in the developing countries. A significant increase in food production is necessary to feed a world population of 9.1 billion people in 2050; the estimated increase in food production is about 70% between 2005 and 2050. Notably, the main increase in food production is expected in the developing countries. Annual production of cereals, for instance, would have to grow by almost one billion tonnes, with 3/4 of this increase occurring in the developing countries [1].

Plant pathogens cause significant crop losses, leading to economic and social disasters. Human practices such as monoculture farming and global trade contribute to the spread of plant pathogens and the emergence of new diseases. Therefore, early detection and identification of pathogens are paramount to reducing associated agricultural losses. Currently, plant pathogens pose a serious threat to the agricultural industry. Up to 40% of economically important crops are lost annually due to phytopathogens and pests [2,3].

Fungi of the genus *Fusarium* constituted the second largest group of pathogens found on seeds. These fungi cause damage to crops and produce toxins that are dangerous for human and animal health. Mycotoxins are a group of compounds originating from the metabolism of filamentous fungi that cause many diseases. Fungi infect different plants, contaminating food products with mycotoxins. More than 300 mycotoxins have been found to induce toxicological effects in mammals. The main mycotoxins with toxic effects include aflatoxins, fumonisins, ochratoxins, patulins, and trichothecenes. Trichothecenes include type A toxins (T-2 toxin, HT-2 toxin), type B toxins (nivalenol, deoxynivalenol), and several others that are produced by a set of Fusarium species, e.g., *Fusarium moniliforme*, *F. equiseti*, *F. culmorum*, *F. solani*, *F. avenaceum*, *F. roseum*, *F. nivale*, *F. tricinctum*, *F. poae*, *F. sporotrichiella*, *F. graminearum*, and *F. sporotrichioides* [4]. The most toxic mycotoxins have a tetracyclic sesquiterpenoid 12,13-epoxytrichothec-9-ene ring and the 12,13-epoxy ring, which is responsible for the toxicological activity. Among the others, T-2 toxin is one of the most dangerous mycotoxins because systemic toxicity can result from any route of exposure, i.e., dermal, oral, or respiratory [5]. The primary target of T-2 toxin and deoxynivalenol (Figure 1) is the A-site of the peptidyl transferase center of the ribosome [6].

Several trichothecenes are under strict control in food and feed due to the risk of mycotoxin contamination, which can occur both in the field and during storage [9]. The European Union regulations set the safe limit for mycotoxin concentrations in grain at 50–200 ppb, while Russian regulations limit it to 0.7 µg/g [10].

Analytical methods for T-2 and HT-2 are well established, and quantification is primarily performed using liquid chromatography coupled with mass spectrometry (LC/MS). For rapid screening, immunochemical methods are used, but these have the disadvantage of cross-reactivity. The existing immunochemical methods have not been validated in interlaboratory studies [11]. LC/MS offers a wide calibration range for trichothecenes, from 0.1 to 200 ng/mL. The technique uses column preconcentration, providing a mean recovery of approx. 50–60%, a limit of detection (LoD) of 0.9 ng/g, and a limit of quantification of 2.9 ng/g [12].

Aptamers are artificial, structured DNA or RNA oligonucleotides that can bind a chemical or biological target with high affinity and specificity. Aptamers are widely used as recognizing elements in biosensors, so-called aptasensors [13,14,15,16,17]. Toxins are also receiving much attention, including marine toxins [18] and mycotoxins [19]. In particular, aptamers to T-2 were developed and applied in graphene oxide-based aptasensors, providing the LoD of 0.4 μM (190 ng/g). The structure of one of the aptamers, Seq.16 aptamer, is shown in Figure 1. This DNA aptamer has a structure with two hairpins that bind T-2 with K_D_ of 20 nM [7].

Here we tried to combine the developed aptamer, namely aptamer Seq.16, with surface-enhanced Raman spectroscopy (SERS), aiming to combine aptamer specificity and SERS sensitivity. SERS is a highly sensitive and robust technique that provides a unique molecular fingerprint with a typical surface-assessed signal intensity increase by 10^6^–10^8^ times [20,21]. SERS is ultimately blended with aptamers, as these recognizing molecules are small and can be accessorized with thiol or amine groups for surface functionalization. Aptamers are more preferable for SERS-based compared to antibodies, as nucleic acids do not quench enhancement of SERS substrates, whereas proteins decrease SERS intensity of the labels by several orders [22]. Moreover, resonant dyes can be incorporated into aptamer structures to achieve even more intense Raman spectra due to the SERRS (surface-enhanced resonant Raman spectroscopy) effect. SERRS occurs when the dye absorbs the same wavelength as the laser used in the Raman spectrometer. The efficiency of dye-plasmon interaction increases by several times, providing intense Raman spectra at extremely low concentrations of the dye [23]. Recently, we have proposed aptasensors with fluorescent Raman active dyes that exhibited changes in intensity upon target binding [24,25]. The same approach is proposed for the detection of low molecular mycotoxins T-2 and DON in wheat grains.

## 2. Results

### 2.1. Optimization of the Aptamer for SERS Aptasensor

Aptamer Seq.16 developed by Chen et al. [7] was adapted for our study to exploit the target-dependent change of SERRS signal of the labeled aptamer. As illustrated in Figure 2, the aptamer Seq.16 was truncated. A series of modifications were proposed (Table 1). The 5′-end was modified with a thiol group to enable functionalization of metal nanoparticles with the aptamer. The 3′-end was modified with various fluorophores (Cyanine-3 (Cy3), Cyanine-5 (Cy5), 5-Carboxytetramethylrhodamine (TAMRA), 5-Carboxyfluorescein (FAM), Rhodamine 6G (R6G)) or dark quenchers (Black Hole Quencher 1 (BHQ1), Black Hole Quencher 2 (BHQ2), Black Hole Quencher 3 (BHQ3), Real Time Quencher 1 (RTQ1)).

The affinity of a set of new aptamers to DON was investigated. While the initial aptamer’s affinity to DON had not been previously estimated, the structural similarity between DON and T-2 made DON a suitable choice for the affinity experiment. DON was chosen for the affinity experiment due to three hydroxyl groups that allow immobilization of the target on the sensor surface. The toxin was conjugated with carboxyl groups of the amino-reactive sensor. The binding of soluble aptamers was assessed using biolayer interferometry (Table 2, Appendix A). The modifications had a significant impact, with dissociation constants (K_D_) ranging from 7 nM to the complete absence of the affinity. This effect could be attributed to the proximity of the modifier to the target binding site in the aptamer. The modifier may form additional non-covalent interactions with the target, leading to increased affinity. The highest affinity was observed with the SH-Fus-Cy3 aptamer (K_D_ = 7.2 nM), which is 20 times higher than the prototype aptamer, SH-Seq.16-Cy3, with K_D_ = 150 nM.

The thermal stability of the modified aptamers was studied to exclude the distortion of the hairpin structure of the aptamer. UV spectroscopy melting experiment revealed minimal if any destabilization of the hairpin structure. Melting temperatures were nearly the same (~57 °C) for aptamers SH-Fus, SH-Fus-Cy3, SH-Fus-Cy5, and SH-Fus-FAM aptamers (Table 3, Appendix A). The SH-Fus-TAMRA aptamer exhibited a slightly lower melting temperature of 1 °C, while other derivatives had melting temperatures in the range of 60–65 °C. Thus, all the aptamers are stable, and the modifications are non-destructive for aptamer secondary structure.

The thermal stability of SH-Fus-Cy3 aptamer was estimated in different salts that are significant for SERS intensity, namely, in chlorides, nitrates, and nitrates with Ca^2+^ addition (Appendix A). The addition of Ca^2+^ ions provided the highest impact on the melting curve (Appendix A and Table 4). The addition of DON decreased melting temperature slightly to 57.8 °C (Appendix A, Table 4).

Next, SERS spectra were recorded with a Raman spectrometer equipped with a laser wavelength of 532 nm. Colloidal silver nanoparticles provide SERS spectra only after aggregation with a buffer. Buffer A provided SERS spectra of SH-Fus-TAMRA, SH-Fus-Cy3, and SH-Fus-FAM. Ca^2+^-containing Buffer B provided much more intensive SERS spectra; spectra of seven different dyes can be detected (Figure 3, Table 5). The SERS intensity was increased 3–100 times.

The aptamer SH-Fus-Cy3 is a preferable recognizing element for the aptasensor due to its high affinity and slow dissociation rate of the complex with the target (Table 2), thermal stability, and the highest SERS intensity. The affinity was retained in Buffer B (K_D_ = 6.0 ± 0.6 nM), facilitating further implementation of the aptamer using any of the studied buffers.

### 2.2. Aptasensor Setup

Silver nanoparticles were incubated with aptamer SH-Fus-Cy3 for 30 min. The functionalized nanoparticles were used immediately after the preparation. Silver nanoparticles were aggregated to provide SERS spectra. Purified toxins (T-2 and DON) were incubated with aptamer-functionalized nanoparticles before aggregation. The addition of toxins led to the increase of SERS spectra of the Cy3 resonant Raman dye (Figure 4). The limit of detection (LoDs) of aptasensor for both toxins were 70 ng/mL (150 nM) in Buffer A and 170 ng/mL (360 nM) in Buffer B. Buffer A is more preferable due to the lower LoD. On the other hand, Buffer B provided a three-fold increase in SERS intensity.

Next, the pure cultures of fungi that produce T-2 and DON toxins were studied. Samples of *Fusarium sporotrichioides* filaments, which produce T-2 toxin, were extracted with 80% *i*-propanol using either heating or ultrasound treatment. Other target and control fungi filaments, including *Fusarium oxysporum*, were prepared similarly. The extraction procedure was tested with these pure cultures. Both target and non-target (off-target) fungi exhibited increased SERS intensity upon silver nanoparticle aggregation with Buffers A and B (Figure 5). DON-producing fungi (*Fusarium culmorum* and *graminearum* [26]) generated the highest signals, suggesting preferential detection of DON. T-2-producing fungi (*Fusarium sporotrichioides* [26]) with the highest T-2 content were also detected with statistically significant differences from the off-target fungi. *Fusarium acuminatum*, known for its alternative production of DON [27], was also detected, unlike other fungi that do not produce DON.

The normalization SERS (target fungi)/SERS (off-target fungi) was introduced to exclude the effect of matrix on the SERS intensity. The concentration dependencies are shown in Figure 5B,C. Heating in the 80% *i*-propanol allows determining 25 µg of *Fusarium sporotrichioides* in the probe. Ultrasound treatment in the 80% *i*-propanol allows determining of 1.9 µg of the target fungi in the probe. The latest protocol was used for the determination of wheat grains.

### 2.3. Determination of Fusarium Graminearum in Wheat Grains

Several samples of wheat grains were prepared. A part of the grains was infected with *Fusarium graminearum*, which produces T-2 and DON toxins. Another part was infected with a control fungus, *Alternaria alternata,* that does not produce T-2 and DON toxins. Control samples were treated with the equal volume of the medium without fungi; untreated grains were studied also. The sensor provided determination of the toxins in the very low concentration range, as *i*-propanol-extracted wheat components impaired nanoparticle aggregation, providing no SERS. The estimation of nanoparticle aggregation was conducted visually (the color of the solution was yellow instead of grey when the aggregation was impaired) and by the decrease of the peak at 230 cm^−1^ down to the full disappearance. The aggregation can be increased due to the aggregation by Buffer B instead of Buffer A (Figure 6), as calcium cations enhance aggregation. Nevertheless, the volume of the extract did not abolish nanoparticle aggregation and was limited to 0.05–0.1 µL. The toxins were successfully detected in the extract from 12.5 µg of infected wheat grains per probe. The decrease in nanoparticle aggregation impairs the exact determination of LoD and quantification of toxins in the samples.

To address the nanoparticle aggregation issue, we modified the extraction solvent. Proteins are known to form a “crown” around nanoparticles, hindering their aggregation and decreasing SERS signal by several folds [28,29]. *i*-Propanol, a polar hydrophilic solvent, was revealed not to provide an efficient separation of proteins and other hydrophilic macromolecules from low molecular toxins. We selected acetonitrile, another polar solvent without donors or acceptors of hydrogen bonds. Acetonitrile is commonly used to precipitate proteins from the samples [30], making it well-suited for our requirements, especially considering the solubility of T-2 and DON toxins in the acetonitrile. The extraction with acetonitrile was conducted in a similar manner to *i*-propanol. Even with the extract volume 100 times larger than in the previous setup, nanoparticle aggregation was not impaired. The sensor demonstrated good specificity, showing a decrease in SERS signal, whereas the uninfected grains and grains with *Alternaria alternata* did not decrease the SERS signal (Figure 7A). In addition, pure acetonitrile slightly increased the signal to 6600 ± 300 a.u. The lowest detectable content of infected grains was 750 µg per probe. Next, the mixtures of uninfected grains and grains with *Fusarium graminearum* were prepared. The aptasensor readily detected mixtures with 25–100% of infected grains (Figure 7B).

The limit of detection for acetonitrile extraction is approx. 50 times larger than the LoD of *i*-propanol extraction. However, the decrease in the sensitivity is compensated by the large SERS intensity and low matrix effect. As a result, the quantification of toxins can be provided.

Additional experiments were performed to recheck aptasensor performance. The SERS signal increased in the presence of purified DON or DON-producing fungi, but it decreased in the presence of infected wheat grains. This effect could be attributed to the matrix effect that is rather large in the case of Raman spectroscopy and SERS. Uninfected wheat grains were mixed with DON-producing fungus to test this hypothesis directly. The quantity of wheat grains was fixed, whereas the quantity of *Fusarium culmorum* was varied. The dependence of the SERS signal on the quantity of DON-producing fungus was monotonously declining with LoD of 1.7 mg/g. Thus, the technique detected 0.17% content of the fungi in the wheat grains. The declining concentration dependence was repeated, supporting the matrix effect as the main cause of the differences in aptasensor performance with purified fungi and infected grains.

### 2.4. A Comparison with Common Assays and Other Biosensors

Liquid chromatography coupled with mass spectrometry (LC/MS) is a common way for determining T-2 and DON in real samples. LC/MS uses column preconcentration, offering a calibration range of 0.1 to 200 ng/mL, an average recovery of approx. 50–60%, a limit of detection (LoD) of 0.9 ng/g, and a limit of quantification of 2.9 ng/g [12]. The primary drawback of LC/MS is the requirement for a well-equipped laboratory and several hours for analysis. Rapid point-of-care tests are highly sought after by agricultural companies. To date, no solution with sufficient specificity and simplicity is commercially available. New techniques are currently under development. Several reviews describe recent works in detail [31,32]. Here, we discuss biosensors for T-2 and DON determination tested with the real samples, as many excellent techniques often yield unsatisfactory results in everyday practice when applied to samples that deviate from purity.

One of the most robust biosensors utilizes labeled DNA aptamers that bind either DON or a complementary DNA strand. DON binding causes the label to approach the SERS-active surface. The detection limit was as low as 0.08 ng/mL under ‘ideal conditions’, while the lowest DON concentration in wheat flour was 5 ng/g [33]. These results are quite close to the robustness of LC/MS, although the sensor has not been tested with *Fusarium*-infected wheat grains. The main disadvantage of this variant is a sophisticated SERS substrate assembled from ZIF-8 (a kind of metalorganic framework) coated with a polydopamine layer and silver nanoparticles. Such sophisticated SERS substrates often provide excellent characteristics within one batch; however, batch-to-batch reproducibility is a rather complicated task. Also, a portable SERS spectrometer has not been tested with this technique.

Li et al. [34] described a competitive immunoassay using biolayer interferometry. The limit of detection for T-2 in barley flour was as low as 20 ng/g on a par with 10 min for an analysis. The technique requires laboratory equipment (Octet Red), which is not a point-of-care device. Subak et al. [35] proposed an aptamer-based electrochemical detection with LOD of 6 ng/g of DON in maize flour within a 1 h detection time. Electrochemical detection has been miniaturized recently, making this variant a promising candidate for practical implementation.

Wu et al. [36] published a polyethyleneimine-functionalized porous reduced graphene oxide-loaded gold nanowire electrochemical sensor capable of detecting DON in corns. This sophisticated sensor enabled direct determination of DON content as low as 0.1% in maize samples. The same performance (0.1% of DON content in cornmeal samples) was achieved using an electrochemical aptasensor based on exonuclease III-mediated signal amplification [37]. As low as 5 × 10^−6^% of DON content was detected using surface plasmon resonance-based technique [38]. The initial two examples provided higher LoD due to their novel techniques, capable of identifying as low as 0.17% content of the fungi in the wheat grains, and DON content is several orders lower than the whole fungi mass. At the same time, these two electrochemical sensors use a sophisticated setup. The novel sensor is much simpler. The surface plasmon resonance-based technique provided the excellent LoDs, their equipment is non-portable, being a part of a highly equipped laboratory.

The proposed sensor supplements a series of biosensors for mycotoxin determination. Moreover, its LoDs may not be the lowest, but it boasts several advantages, including a portable SERS spectrometer, inexpensive SERS substrates, and rapid analysis. The sensor also exhibits exceptional specificity, being able to distinguish wheat grains infected with T-2/DON-producing *Fusarium* strains from T-2/DON-free fungi. These features make our novel aptasensor a promising candidate for broader applications beyond highly equipped laboratories.

## 3. Materials and Methods

### 3.1. Reagents

Inorganic salts and tris were purchased from AppliChem GmbH (Darmstadt, Germany). The reagents from Sigma-Aldrich (St. Louis, MO, USA) were used for nanoparticle synthesis, including silver nitrate (AgNO_3_) and hydroxylamine hydrochloride (NH_2_OH-HCl) of the highest purity available. N-cycloehexyl-N-(2-morpholinoethyl)-carbodiimide-methyl-p-toluolsulfonat (CME-CDI) from Chem. Fabric (Karlsruhe, Germany) and sodium salt of N-hydroxysulfosuccinimide (s-NHS) from Chem-Impex Int’l (Wood Dale, IL, USA) were used. Standard solutions of T2 and DON toxins in acetonitrile with a concentration of 100 µg/mL were purchased from the All-Russian Research Institute of Veterinary Sanitary, Hygiene and Ecology (Moscow, Russia). Modified oligonucleotides were synthesized by Synthol (Moscow, Russia). The sequences are provided in Table 1. All solutions were prepared using ultrapure water produced with Millipore (Merck Millipore, Burlington, MA, USA).

### 3.2. Assembly of Aptamers

The assembly of the aptamer structure was conducted according to the following algorithm. The aptamers were prepared in Buffer A (10 mM Tris-HCl pH 7.3, 140 mM NaNO_3_, and 10 mM KNO_3_), Buffer B (10 mM Tris-HCl pH 7.3, 140 mM NaNO_3_, 10 mM KNO_3_, 5 mM CaCl_2_), or Buffer C (10 mM Tris-HCl pH 7.3, 140 mM NaCl, and 10 mM KCl). The aptamer concentration was 20 µM for SERS experiments and 1 µM for affinity assay and UV melting experiments. The solutions were heated to 95 °C, maintained at that temperature for 5 min, and then cooled to room temperature.

### 3.3. UV Melting Experiments

Aptamer solution with a concentration of 1 μM in Buffer A, B, or C was placed in quartz cuvettes with a path of 1 cm. UV spectra were acquired using a Hitachi U2900 spectrophotometer (Hitachi, Tokio, Japan) equipped with a thermoelectric temperature regulator. The spectra were acquired at the wavelength 260 nm. The spectrum of the buffer was subtracted as a baseline. The samples were heated with a mean ramp of 1.0 °C/min. The melting experiments were conducted in the range of 20–85 °C. The melting temperatures were derived from the temperature dependencies of UV absorption at 260 nm.

### 3.4. Affinity Assay

The affinity of the aptamers to mycotoxin (DON) was estimated using biolayer interferometry (Blitz, ForteBIO, Menlo Park, CA, USA). The experiments were conducted at 20 °C. Samples were placed in black 0.5 mL tubes (Sigma-Aldrich, St. Louis, MO, USA) in a 220 μL volume. Biosensors intended for the amine coupling reaction (Octet AR2G biosensors, ForteBio, Menlo Park, CA, USA) were hydrated for 10 min in water. The sensors were activated for 5 min in the solution of 200 mM CME-CDI and 100 mM s-NHS. Then DON was loaded from a 10 µL drop of 20–100 µg/mL solution. The sensor was washed and blocked with 100 mM HEPES-HCl buffer with pH 7.5 for 3 min. After the signal stabilization in Buffer A, the association step was conducted with aptamer solution in concentrations of 1000, 500, 250, or 125 nM in Buffer A. Then the dissociation step was conducted in the same buffer. Both association and dissociation stages were monitored for 200 s.

### 3.5. Preparation of Silver Nanoparticles

Preparation of the silver colloids by reducing a silver nitrate solution with hydroxylamine hydrochloride was conducted according to the method of Leopold and Lendl [39]. The silver nanoparticles were prepared at room temperature under vigorous stirring by dropwise addition of a 10 mM solution of AgNO_3_ (10 mL) to a mixture of 1.67 mM NH_2_OH-HCl with 3.3 mM NaOH (90 mL). The solution was kept under stirring for 1 h after addition of silver nitrate to ensure a reproducible preparation protocol. A light brown-colored colloidal suspension was obtained with a final pH of 7.0. Before usage of silver nanoparticles for SERS experiments, they were left for 3 months in a refrigerator at +6–+8 °C for aging. The nanoparticles were characterized with absorption spectroscopy, scanning electron microscopy, dynamic light scattering, and ζ-potential according to the previous work [29].

### 3.6. Preparation of Samples with Fusarium Fungi

Twelve strains of various species of fungi of the genus *Fusarium* were isolated from plant material collected in different regions of the Russian Federation. The biomaterial (roots, leaves, stems, spikelet scales) was washed with running water, dried in air, and crushed with scissors into fragments 2–4 mm long. Then, using tweezers, each fragment was dipped into 96% ethanol for 3 s and flamed (quickly burned in the flame of an alcohol burner), after which it was placed on the surface of potato-glucose agar with gentamicin in standard glass Petri dishes (100 fragments, 4 fragments per cup). The cups were incubated in a thermostat at 26–28 °C for 7–10 days (before the appearance of colonies); then, the colonies were transplanted to a fresh nutrient medium of potato-glucose agar, grown, micro-preparations were prepared, viewed under a microscope, and the taxonomic affiliation of microorganisms was determined.

### 3.7. Fusarium Strains

*Fusarium acuminatum* TB-20-5 strain was isolated from wheat stalks from the Tambov region of the Central Chernozem region of the Russian Federation in 2020. *Fusarium avenaceum* SKZ-5 strain was isolated from wheat spikelets from Karachay-Cherkessia in the North Caucasus region of the Russian Federation in 2023. *Fusarium culmorum* MO-3-2011 strain was isolated from barley roots from the Moscow region, the Central region of the Russian Federation, in 2011. *Fusarium equiseti* strain Kr-20-51 was isolated from wheat leaves from the Krasnodar Territory of the North Caucasus region in 2020. *Fusarium graminearum* FG-30 strain was isolated from wheat spikelets from the Moscow region of the Central Region of the Russian Federation in 2017. *Fusarium poae* strain Kr-20-14 was isolated from wheat roots from the Krasnodar Territory of the North Caucasus region of the Russian Federation in 2020. *Fusarium proliferatum* 329-1 strain was isolated from tomato stalks from the Leningrad region of the Northwestern region of the Russian Federation in 2021. *Fusarium roseum (sambucinum)* Ct-20-3 strain was isolated from wheat leaves from the Amur region of the Far Eastern region of the Russian Federation in 2020. *Fusarium sporotrichioides* Ct-20-8st strain was isolated from wheat stalks from the Amur region of the Far Eastern region of the Russian Federation in 2020. *Fusarium tricinctum* TB-20-4 strain was isolated from wheat stalks from the Tambov region of the Central Chernozem region of the Russian Federation in 2020. *Fusarium verticilloides* FV-2018 strain was isolated from wheat roots from the Moscow region, the Central region of the Russian Federation, in 2018.

Furthermore, 10 mg of *Fusarium* filaments were incubated in 250 µL of 80% *i*-propanol with subsequent incubation at 65 °C for 5 min, producing a 40 mg/mL solution. Alternatively, the same solution was subjected to ultrasound treatment with Sapphire (Sapphire LLC., Moscow, Russia) for 5 min at 37 °C.

### 3.8. Wheat Grains Infected with Fusarium Fungi

The Alekseich wheat variety is a medium-late, high-yielding, short-stemmed variety with a growing season of 214–306 days. The wheat grain raw material was produced by AgroMir-Sids (Krasnodar Territory, Russia) using the seed library from P.P. Lukyanenko National Grain Center (Krasnodar, Russia). This wheat variety is resistant to brown rust, yellow rust, stem rust, and powdery mildew. It is moderately susceptible to *Fusarium* and *Septoria*. The concrete samples were obtained from im.Kirov LLC. (Rostov region, Russia); the harvest of the year 2022 was used.

The process of infection of wheat grains with *Fusarium* was as follows. The grains were weighed and packaged in vacuum bags in amounts of 7.0 g per bag. Two packages were allocated for dry and wet control samples, whereas other packages were infected with various pathogens. Bags were dipped in boiling water for 40 min in order to provide disinfection. Then the bags were cooled at room temperature. Glass Petri dishes were used after washing with distilled water, wiping with alcohol, and drying. Wheat grains from different packages were transferred to separate Petri dishes. A dish with a dry sample was immediately sealed with paraphylum. Three milliliters of distilled water was injected into a dish with a wet sample with a Pasteur pipette, and then the dish was sealed. Parts of the *Fusarium* mycelium of the same size (a sector the size of one quarter of a cup) were selected with a disposable scraper and transferred to separate sterile vials with screw caps. Then 9 mL of distilled water was added to these vials with a Pasteur pipette. The resulting solution was shaken for 3 min to better saturate the pathogen spores. Then 3 mL of this mixture was transferred to Petri dishes with grains. The dishes were then sealed with a paraphylum. The dishes were packed in airtight bags and stored in a thermostat at a temperature of 28 °C.

The grains were ground with a coffee grinder. A sample of 250 mg of wheat flour was mixed with 1 mL of *i*-propanol. Alternatively, a sample of 100 mg of wheat flour was mixed with 0.4 mL of acetonitrile. The mixtures were subjected to ultrasound treatment with Sapphire (Sapphire LLC., Moscow, Russia) for 10 min at 37 °C. After centrifugation, the supernatants were collected.

### 3.9. Aptasensor Maintenance

The aptamer-modified nanoparticles were prepared prior to the experiment. An aliquot of thiolated aptamer in Buffer A was diluted in nanoparticle solution to a final concentration of aptamer of 20 nM, followed by 30 min of incubation. An aliquot of purified toxins or biological samples (0.5–20 µL) was added to 200 µL of aptamer-modified nanoparticles. After 10 min of incubation, the nanoparticles were aggregated by 300 µL of Buffer A or B. The SERS spectra were recorded after 1 min of incubation. The exposure time was 100 ms with 50 repeats. The handheld Raman analyzer RaPort (Enhanced Spectrometry, Meridian, MS, USA) with a laser excitation wavelength of 532 nm and a power of 30 mW was used.

## Figures and Tables

**Figure 1 ijms-25-09534-f001:**
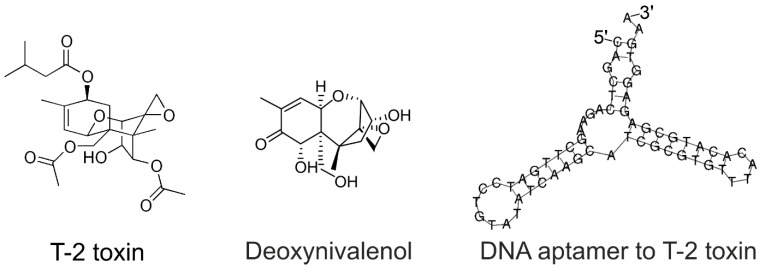
Structures of T-2 toxin, deoxynivalenol, and DNA aptamer selected to T-2 toxin by Chen et al. [7]. The secondary structure of the aptamer was built using RNAfold [8].

**Figure 2 ijms-25-09534-f002:**
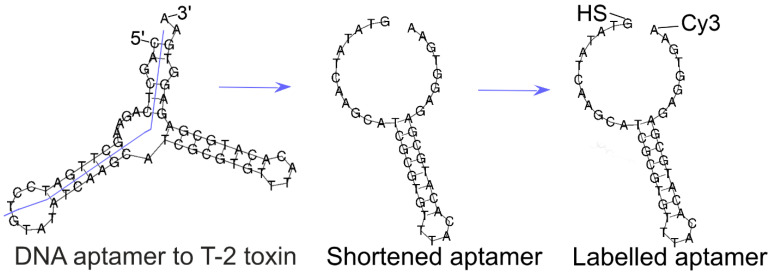
Optimization of the secondary structure of the aptamer to T-2 toxin. Twenty nucleotides were truncated from aptamer Seq.16, providing aptamer Fus. Aptamer Fus was modified with Raman dye (Cyanine-3, Cy3) at the 3′-end as well as with thiol group at the 5′-end.

**Figure 3 ijms-25-09534-f003:**
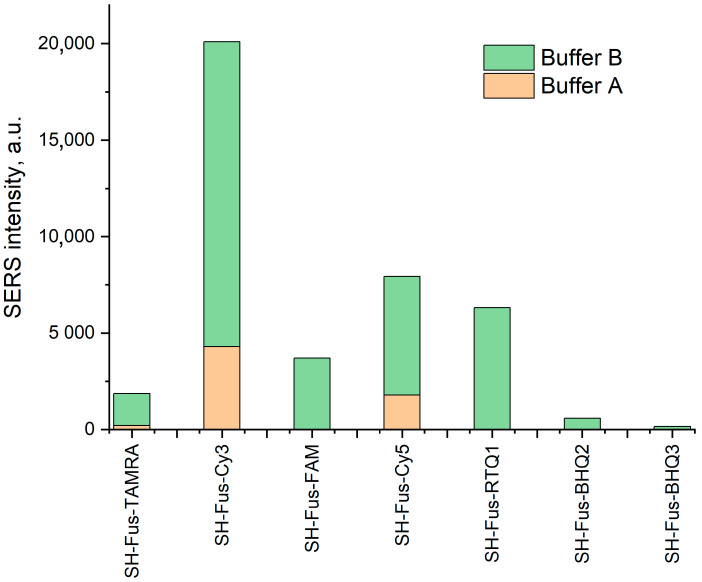
Comparison of SERS intensity of the different dyes in Buffers A and B.

**Figure 4 ijms-25-09534-f004:**
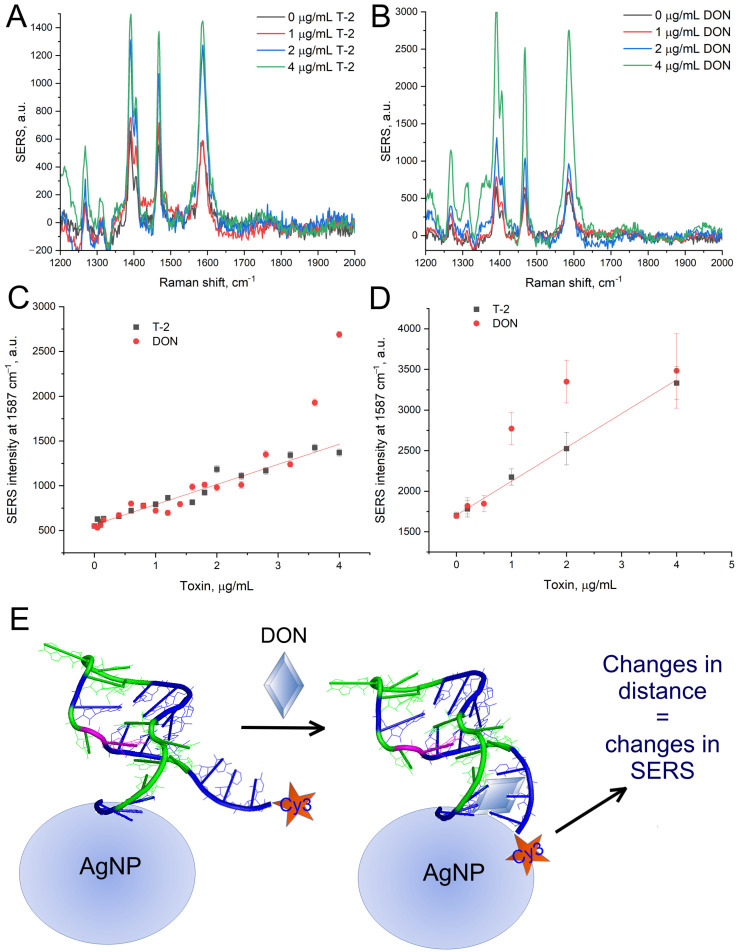
Performance of the aptasensor with purified toxins. SERS spectra of aptamer SH-Fus-Cy3 in the presence of T-2 (**A**) and DON (**B**) in Buffer A. The dependence of SERS intensity on T-2 and DON concentration in Buffers A (**C**) and B (**D**). A schematic explanation of the reason for SERS intensity changes is shown (**E**). The changes in distance between cyanine 3 and silver nanoparticles affect SERS intensity of the cyanine 3 bands in spectra.

**Figure 5 ijms-25-09534-f005:**
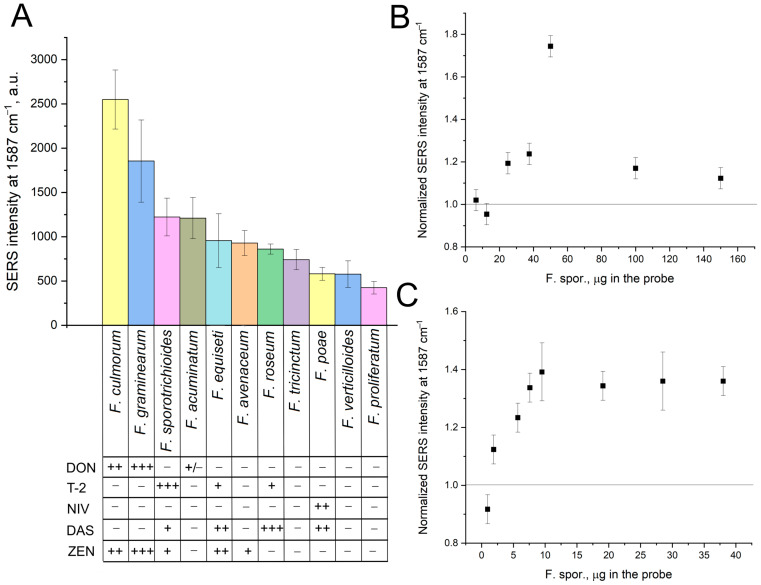
Optimization of extraction procedure with pure cultures of *Fusarium* fungi. The dependence of normalized SERS intensity on the content of *Fusarium* fungi added. Comparison of target and off-target fungi with the content of 20 µg per probe; the relative contents of toxins are shown as +++ (high), ++ (medium), + (low), and − (absent) (**A**). Heating procedure (**B**) and ultrasound procedure (**C**) are compared with normalization of SERS signal of *Fusarium sporotrichioides* to the signal of *Fusarium oxysporum*. Buffer A was used for concentration dependencies; Buffer B was used for a comparison of the target and off-target fungi.

**Figure 6 ijms-25-09534-f006:**
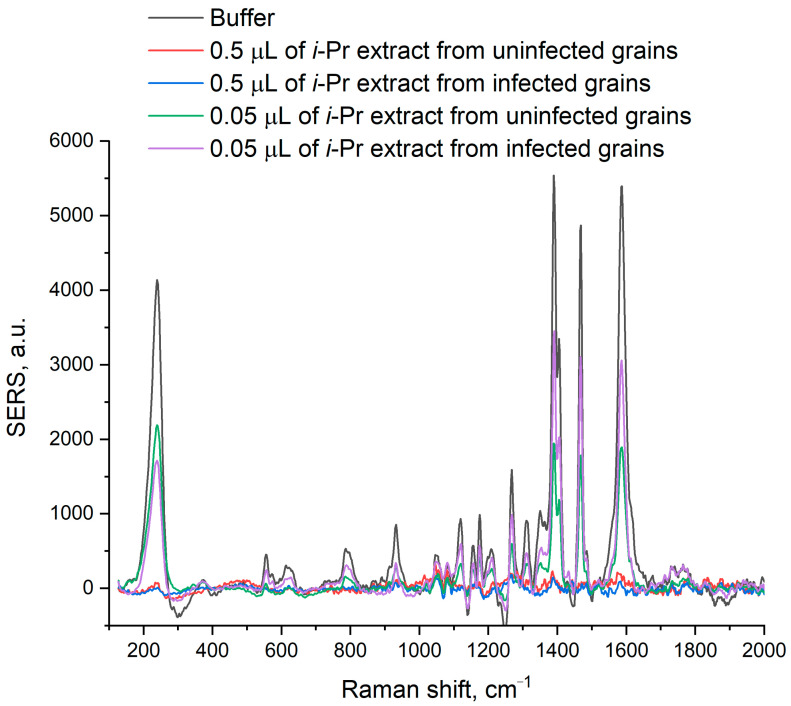
SERS spectra of isopropanol extracts of uninfected wheat grains and grains infected with *Fusarium graminearum*. The components of the extracts disrupt nanoparticle aggregation in concentration-dependence manner.

**Figure 7 ijms-25-09534-f007:**
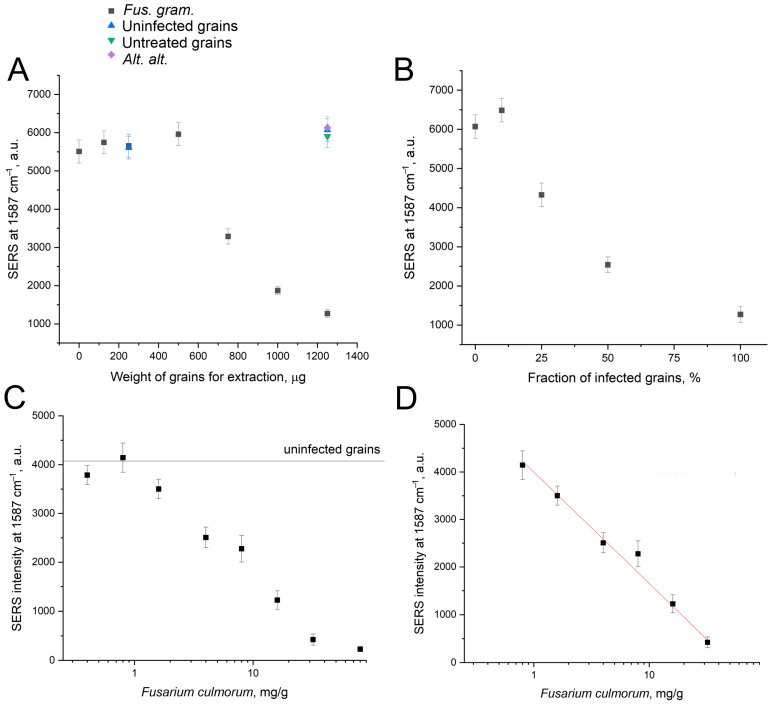
The robustness of the aptasensor in determination of wheat grains infected with *Fusarium graminearum*. The dependence of the SERS signal of the sensor on the amount of wheat grains subjected to extraction with acetonitrile (**A**). The fraction of the *Fusarium graminearum*-infected grains in the mix with uninfected grains (**B**). *Fus. gram*. is *Fusarium graminearum*; *Alt. alt*. is *Alternaria alternata*. Uninfected grains were treated with a buffer. An artificial mixture of uninfected wheat grains with *Fusarium culmorum* was also studied (**C**); the linearization of the dependence is shown in subset (**D**). The fungus concentration is shown in mg per 1 g of wheat grains.

**Table 1 ijms-25-09534-t001:** Sequences of DNA oligonucleotides used in the study.

Code	Sequence (5→3)
SH-Seq.16	(SH-(CH_2_)_6_)-CAGCTCAGAAGCTTGATCCTGTATATCAAGC ATCGCGTGTTTACACATGCGAGAGGTGAA
SH-Seq.16-Cy3	(SH-(CH_2_)_6_)-CAGCTCAGAAGCTTGATCCTGTATATCAAGC ATCGCGTGTTTACACATGCGAGAGGTGAA-(Cy3)
SH-Fus	(SH-(CH_2_)_6_)-GTATATCAAGCATCGCGTGTTTACACATGCG AGAGGTGAA
SH-Fus-BHQ1	(SH-(CH_2_)_6_)-GTATATCAAGCATCGCGTGTTTACACATGCG AGAGGTGAA-(BHQ1)
SH-Fus-BHQ2	(SH-(CH_2_)_6_)-GTATATCAAGCATCGCGTGTTTACACATGCG AGAGGTGAA-(BHQ2)
SH-Fus-BHQ3	(SH-(CH_2_)_6_)-GTATATCAAGCATCGCGTGTTTACACATGCG AGAGGTGAA-(BHQ3)
SH-Fus-Cy3	(SH-(CH_2_)_6_)-GTATATCAAGCATCGCGTGTTTACACATGCG AGAGGTGAA-(Cy3)
SH-Fus-Cy5	(SH-(CH_2_)_6_)-GTATATCAAGCATCGCGTGTTTACACATGCG AGAGGTGAA-(Cy5)
SH-Fus-FAM	(SH-(CH_2_)_6_)-GTATATCAAGCATCGCGTGTTTACACATGCG AGAGGTGAA-(FAM)
SH-Fus-R6G	(SH-(CH_2_)_6_)-GTATATCAAGCATCGCGTGTTTACACATGCG AGAGGTGAA-(R6G)
SH-Fus-RTQ1	(SH-(CH_2_)_6_)-GTATATCAAGCATCGCGTGTTTACACATGCG AGAGGTGAA-(RTQ1)
SH-Fus-TAMRA	(SH-(CH_2_)_6_)-GTATATCAAGCATCGCGTGTTTACACATGCG AGAGGTGAA-(TAMRA)
SH-Hairpin-Cy3	(SH-(CH_2_)_6_)-TGGCAATGTTGACTTCCTCAAGGAACCA-(Cy3)

**Table 2 ijms-25-09534-t002:** Affinity of the Fus aptamer modified with fluorophores or fluorescent quenchers at the 3′-end as well as with thiol group at the 5′-end. Equilibrium dissociation constants (K_D_), association (k_ass_), and dissociation (k_diss_) rate constants are provided. The complexes were assembled in Buffer C. n.d. means no detectable binding.

Aptamer	K_D_, nM	k_ass_, µM^−1^ s^−1^	k_diss_, s^−1^
SH-Seq.16-Cy3	150 ± 40	0.17 ± 0.08	0.026 ± 0.007
SH-Fus	290 ± 80	0.011 ± 0.005	0.0031 ± 0.0011
SH-Fus-BHQ1	64 ± 16	0.12 ± 0.04	0.007 ± 0.003
SH-Fus-BHQ2	120 ± 40	0.08 ± 0.04	0.010 ± 0.003
SH-Fus-BHQ3	>1000	n.d.	n.d.
SH-Fus-Cy3	7.2 ± 1.5	0.24 ± 0.09	0.0017 ± 0.0003
SH-Fus-Cy5	24 ± 5	0.19 ± 0.07	0.0047 ± 0.0007
SH-Fus-FAM	>1000	n.d.	n.d.
SH-Fus-R6G	140 ± 30	0.024 ± 0.09	0.0034 ± 0.0007
SH-Fus-RTQ1	190 ± 40	0.018 ± 0.007	0.0033 ± 0.0007
SH-Fus-TAMRA	>1000	n.d.	n.d.

**Table 3 ijms-25-09534-t003:** Thermal stability of the Fus aptamer modified with fluorophores and fluorescent quenchers at the 3′-end as well as with thiol group at the 5′-end. Melting temperatures were derived from UV melting experiments. The aptamers were assembled in Buffer C.

Aptamer	Melting Temperature, °C
SH-Fus	57.5 ± 0.1
SH-Fus-BHQ1	63.6 ± 0.1
SH-Fus-BHQ2	63.3 ± 0.1
SH-Fus-BHQ3	60.5 ± 0.1
SH-Fus-Cy3	57.3 ± 0.1
SH-Fus-Cy5	56.7 ± 0.1
SH-Fus-FAM	57.2 ± 0.1
SH-Fus-R6G	61.9 ± 0.1
SH-Fus-RTQ1	65.0 ± 0.1
SH-Fus-TAMRA	55.8 ± 0.1

**Table 4 ijms-25-09534-t004:** Thermal stability of the Fus aptamer with Cyanine-3 at the 3′-end as well as with thiol group at the 5′-end. Melting temperatures were derived from UV melting experiments. The aptamer was assembled in different buffers with or without target substance (DON).

Aptamer	Melting Temperature, °C
Buffer C	57.3 ± 0.1
Buffer B	62.5 ± 0.1
Buffer B with 1.5 vol./vol. % AcCN	59.9 ± 0.1
Buffer B with 1 equivalent of DON and 1.5 vol./vol. % AcCN	57.8 ± 0.1

**Table 5 ijms-25-09534-t005:** Enhancement of SERS intensity of different dyes in the presence of calcium cations. The exposure time was 300 ms. n.d. means no detectable signal.

Aptamer	Buffer A	Buffer B	Ratio Buffer B/Buffer A
SH-Fus-TAMRA	210	1650	7.9
SH-Fus-Cy3	4300	15,800	3.7
SH-Fus-FAM	n.d.	3680	>100
SH-Fus-Cy5	1790	6140	3.4
SH-Fus-RTQ1	n.d.	6300	>100
SH-Fus-BHQ2	n.d.	570	>100
SH-Fus-BHQ3	n.d.	150	>100

## Data Availability

Data is contained within the article and Appendix A.

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
