# Peer review of "A Surface-Enhanced Raman Spectroscopy-Based Aptasensor for the Detection of Deoxynivalenol and T-2 Mycotoxins"

_ijms, 2024, doi:10.3390/ijms25179534_

Round 1

Reviewer 1 Report

Comments and Suggestions for Authors

In this work, the authors reported the development of a ligand sensor based on surface-enhanced Raman spectroscopy (SERS) for the detection of T-2 and deoxynivalenol in wheat grains. The detection rate of pathogenic bacteria in wheat kernel was as low as 0.17%. SERS is eventually mixed with aptamers and can be surface functionalized with mercaptan or amine groups. In addition, resonant dyes provide more of a strong Raman spectrum, providing SERRS (surface-Enhanced Resonant Raman spectroscopy). However, the experimental data provided in this paper are insufficient and some descriptions are unclear. Therefore, it is a pity that this manuscript is not suitable for publication in International Journal of Molecular Sciences, and the detailed reasons are as follows.

1.In this work, the use of hyphens to represent a certain range of values is not consistent, both "-" and "–" are used.

2.The introduction fails to compare the types and performance of traditional ligand sensors, thereby not adequately highlighting the significance of the current work.

3.There is confusion and unclear expression in the introduction. For example, the article mentions the combination of aptamers with surface-enhanced Raman spectroscopy (SERS), but it is unclear whether aptamers were developed first or if SERS technology was studied first. Logically, the development and application of aptamers should be introduced first, and then how to combine them with SERS technology should be explained. SERS and SERRS (Surface-Enhanced Resonance Raman spectroscopy) are mentioned, but the distinction between the two is not elaborated upon. This can lead to confusion among readers, as SERRS is a special form of SERS that usually involves the use of resonant dyes.

4.More detailed data interpretation and theoretical background should be provided to support the observations and conclusions mentioned. For example, it is mentioned that the SH-Fus-Cy3 aptamer has the highest affinity (KD=7.2 nM), which is 20 times that of the prototype aptamer SH-Seq16-Cy3 (KD=150 nM). But the reason for the change was not explained in detail. This change could be attributed to the introduction of modifiers that alter the conformation of the aptamer, the exposure of the active site, or changes in the binding pattern to the target molecule.

5.To address the issue of nanoparticle aggregation, acetonitrile was chosen as the solvent. In the extraction process of nanoparticles, the choice of solvent should be based on its solubility to nanoparticles, extraction efficiency of target substances, and the ability to protect the structure of nanoparticles and other factors. Therefore, there is a lack of more scientific evidence to support the rationality of choosing acetonitrile as a solvent.

6.In this work, there is a lack of discussion on in-depth analysis and interpretation of the results in the context of existing literature.

7.It is pointed out that the further implementation of novel aptamer sensors in well-equipped laboratories is encouraged in the conclusion part, but the practical application prospects have not been prospected.

Comments on the Quality of English Language

The language needs to be improved

Author Response

Q.0. In this work, the authors reported the development of a ligand sensor based on surface-enhanced Raman spectroscopy (SERS) for the detection of T-2 and deoxynivalenol in wheat grains. The detection rate of pathogenic bacteria in wheat kernel was as low as 0.17%. SERS is eventually mixed with aptamers and can be surface functionalized with mercaptan or amine groups. In addition, resonant dyes provide more of a strong Raman spectrum, providing SERRS (surface-Enhanced Resonant Raman spectroscopy). However, the experimental data provided in this paper are insufficient and some descriptions are unclear. Therefore, it is a pity that this manuscript is not suitable for publication in International Journal of Molecular Sciences, and the detailed reasons are as follows.

A.0. Thank you for an attention to our work. Based on the reviewer’s comments a revision has been performed. We agree that some descriptions are unclear. The text was modified. At the same time, we found no comments that require additional experiments or additional Figures/Tables.

Q.1. In this work, the use of hyphens to represent a certain range of values is not consistent, both "-" and "–" are used.

A.1. The use of hyphens was checked; one issue was found and fixed.

Q.2. The introduction fails to compare the types and performance of traditional ligand sensors, thereby not adequately highlighting the significance of the current work.

A.2. Yes, thank you for this comment. A detailed comparison of aptamers and antibodies is out of the scope of this work. However, a short comment is necessary to explain the choice of the aptamers. The introduction was modified in the following way:

‘Aptamers are more preferable for SERS-based compared to antibodies as nucleic acids do not quench enhancement of SERS substrates, whereas proteins decrease SERS intensity of the labels by several orders (Zhang et al., 2010). Additionally, resonant dyes can be incorporated into aptamer structure providing even more intense Raman spectra due to SERRS (surface-enhanced resonant Raman spectroscopy) effect (Litti et al., 2020).’

Q.3. There is confusion and unclear expression in the introduction. For example, the article mentions the combination of aptamers with surface-enhanced Raman spectroscopy (SERS), but it is unclear whether aptamers were developed first or if SERS technology was studied first. Logically, the development and application of aptamers should be introduced first, and then how to combine them with SERS technology should be explained. SERS and SERRS (Surface-Enhanced Resonance Raman spectroscopy) are mentioned, but the distinction between the two is not elaborated upon. This can lead to confusion among readers, as SERRS is a special form of SERS that usually involves the use of resonant dyes.

A.3. We agree with referee’s logic. We followed it during the introduction writing. Aptamers and their application were introduced in lines 78-82; then SERS was introduced as a useful technique for aptasensor development (lines 85-95). This description was modified to enhance this logic. As for difference between SERS and SERRS, it is described in lines 92-95. The description was appended with a short comment to explain the difference.

‘SERRS occurs when the dye can absorb the same part of spectrum that emitted by a laser of Raman spectrometer. The efficiency of dye-plasmon interaction increases by several times providing intense Raman spectra at extremely low concentration of the dye (Litti et al., 2020).’

Q.4. More detailed data interpretation and theoretical background should be provided to support the observations and conclusions mentioned. For example, it is mentioned that the SH-Fus-Cy3 aptamer has the highest affinity (KD=7.2 nM), which is 20 times that of the prototype aptamer SH-Seq16-Cy3 (KD=150 nM). But the reason for the change was not explained in detail. This change could be attributed to the introduction of modifiers that alter the conformation of the aptamer, the exposure of the active site, or changes in the binding pattern to the target molecule.

A.4. Yes, we agree absolutely with this interpretation. We wrote the explanation in the 2nd paragraph of the ‘Results’ section. Namely: ‘…The effect of the modifications was tremendous. The dissociation constants (KD) varied from 7 nM to the complete absence of the affinity. This effect could be attributed to the proximity of the modifier to the target binding site in the aptamer.’

We also add the following phrase to address the reviewer comment (lines 125-126):

‘The modifier can form additional non-covalent interactions with the target providing the increase in affinity.’

We have no convinced structural data up to date to propose an exact explanation. However, we have unpublished data that suggest the binding site to be located near the interface made up by 5’-end, 3’-end and the duplex junction. If the 5’- or 3’-end are included into another hairpin or deleted, the affinity disappears. We plan a detailed study on the aptamer structure, so these first results are not included in the current version.

A.5. To address the issue of nanoparticle aggregation, acetonitrile was chosen as the solvent. In the extraction process of nanoparticles, the choice of solvent should be based on its solubility to nanoparticles, extraction efficiency of target substances, and the ability to protect the structure of nanoparticles and other factors. Therefore, there is a lack of more scientific evidence to support the rationality of choosing acetonitrile as a solvent.

Q.5. Here, a misunderstanding occurred. We did not extract nanoparticles. We extracted toxins from wheat grains. The extracts were added to nanoparticles. As extraction procedure lacks specificity, some other components were extracted on a par with toxins. And these components affect nanoparticle aggregation. We agree that an explanation is useful in this part. We add the following:

‘Proteins are known to be a reason of a crown formation around nanoparticles impairing their aggregation and decreasing SERS by several folds (Mi et al., 2024; Zavyalova et al., 2021). i-Propanol is a polar hydrophilic solvent that was revealed not to provide an efficient separation of proteins and other hydrophilic macromolecules from low molecular toxins. We choose acetonitrile, another polar solvent with no donors or acceptors of hydrogen bonds. Acetonitrile is often used to precipitate proteins from the samples (Polson et al., 2003), so it perfectly fit for requirements.’

Q.6.In this work, there is a lack of discussion on in-depth analysis and interpretation of the results in the context of existing literature.

A.6. As we understood, this comment summarizes the common problem explained in comments 3-5. We elaborate the text according to reviewer’s comments. We think, now the results have better interpretation and connection with the theoretical background.

Q.7. It is pointed out that the further implementation of novel aptamer sensors in well-equipped laboratories is encouraged in the conclusion part, but the practical application prospects have not been prospected.

A.7. In the conclusion part we wrote, that our technique does NOT require well-equipped laboratories. We developed a simple technique that is compatible with portable devices and requires <1 hour. We studied the real samples of wheat grains suggesting sensor robustness. We compared the LoD in the real samples with LoD of other techniques. The ‘Conclusion’ part was renamed and rewritten to compare the technique with others discussing strengths and weaknesses of the different assays from the point of view of possible practical implementation.

Reviewer 2 Report

Comments and Suggestions for Authors

The manuscript focuses on the development of a biosensor for the detection of harmful mycotoxins produced by Fusarium fungi in cereals, specifically wheat grains. These mycotoxins are a major concern for food safety and development of new analytical devices is highly desired. The researchers chosen an approach based on an aptasensor that uses surface-enhanced Raman spectroscopy for detection. The key advantages of this technology are its portability, cost-effectiveness, and rapid analysis time, making it suitable for use outside standard laboratories.

I have not found significant drawbacks in the manuscript, though I have some recommendations. 1. The manuscript lacks full comparison of the results with the current chemosensors and biosensors on mycotoxins. The recent progress should be discussed and compared with the results achieved.

2. The specific advantages and disadvantages compared to standard methods should be also discussed.

3. A figure (an equivalent to graphical abstract) that describes the principle of the assay would improve the attractivity of the manuscript for readers.  

Author Response

The manuscript focuses on the development of a biosensor for the detection of harmful mycotoxins produced by Fusarium fungi in cereals, specifically wheat grains. These mycotoxins are a major concern for food safety and development of new analytical devices is highly desired. The researchers chosen an approach based on an aptasensor that uses surface-enhanced Raman spectroscopy for detection. The key advantages of this technology are its portability, cost-effectiveness, and rapid analysis time, making it suitable for use outside standard laboratories.

I have not found significant drawbacks in the manuscript, though I have some recommendations.

Comment 1. The manuscript lacks full comparison of the results with the current chemosensors and biosensors on mycotoxins. The recent progress should be discussed and compared with the results achieved.

Answer 1. Thank you for this valuable comment. We renamed and rewrote the ‘Conclusion’ part to provide in-depth comparison with other techniques discussing their strengths and weaknesses from the point of view of possible practical implementation.

Comment 2. The specific advantages and disadvantages compared to standard methods should be also discussed.

Answer 2. A comparison with LC/MS was also added to the ‘A comparison with common assays and other biosensors’ subsection.

Comment 3. A figure (an equivalent to graphical abstract) that describes the principle of the assay would improve the attractivity of the manuscript for readers.  

Answer 3. Yes, we agree, that the graphical explanation of the assay is necessary. The scheme was added in Figure 3E.

Round 2

Reviewer 1 Report

Comments and Suggestions for Authors

The author has made the corresponding changes according to the review requests and I agree to accept it for publication in IJMS.

Comments on the Quality of English Language

At the level of English, appropriate revision should be improved.